# Sports-Related Gastrointestinal Disorders: From the Microbiota to the Possible Role of Nutraceuticals, a Narrative Analysis

**DOI:** 10.3390/microorganisms12040804

**Published:** 2024-04-16

**Authors:** Alexander Bertuccioli, Giordano Bruno Zonzini, Massimiliano Cazzaniga, Marco Cardinali, Francesco Di Pierro, Aurora Gregoretti, Nicola Zerbinati, Luigina Guasti, Maria Rosaria Matera, Ilaria Cavecchia, Chiara Maria Palazzi

**Affiliations:** 1Department of Biomolecular Sciences, University of Urbino Carlo Bo, 61122 Urbino, Italy; alexander.bertuccioli@uniurb.it (A.B.); giordano.zonzini@uniurb.it (G.B.Z.); marco.cardinali@uniurb.it (M.C.); 2Microbiota International Clinical Society, 10123 Torino, Italy; maxcazzaniga66@gmail.com (M.C.); f.dipierro@vellejaresearch.com (F.D.P.); auroragregoretti@gmail.com (A.G.); jajamatera74@gmail.com (M.R.M.); ilaria.cavecchia@gmail.com (I.C.); 3Scientific & Research Department, Velleja Research, 20125 Milano, Italy; 4Department of Internal Medicine, Infermi Hospital, AUSL Romagna, 47921 Rimini, Italy; 5Department of Medicine and Surgery, University of Insurbia, 21100 Varese, Italy; nicola.zerbinati@uninsubria.it (N.Z.); luigina.guasti@uninsubria.it (L.G.)

**Keywords:** IBS, microbiota, sport, GIS, nutraceuticals

## Abstract

Intense physical exercise can be related to a significant incidence of gastrointestinal symptoms, with a prevalence documented in the literature above 80%, especially for more intense forms such as running. This is in an initial phase due to the distancing of the flow of blood from the digestive system to the skeletal muscle and thermoregulatory systems, and secondarily to sympathetic nervous activation and hormonal response with alteration of intestinal motility, transit, and nutrient absorption capacity. The sum of these effects results in a localized inflammatory process with disruption of the intestinal microbiota and, in the long term, systemic inflammation. The most frequent early symptoms include abdominal cramps, flatulence, the urge to defecate, rectal bleeding, diarrhea, nausea, vomiting, regurgitation, chest pain, heartburn, and belching. Promoting the stability of the microbiota can contribute to the maintenance of correct intestinal permeability and functionality, with better control of these symptoms. The literature documents various acute and chronic alterations of the microbiota following the practice of different types of activities. Several nutraceuticals can have functional effects on the control of inflammatory dynamics and the stability of the microbiota, exerting both nutraceutical and prebiotic effects. In particular, curcumin, green tea catechins, boswellia, berberine, and cranberry PACs can show functional characteristics in the management of these situations. This narrative review will describe its application potential.

## 1. Introduction

Physical exercise is generally considered among the tools with which it is possible to intervene in the management of physical fitness and health for prevention and treatment purposes, in particular with regard to metabolic and immune health [1,2,3,4]. Endurance exercise includes activities with significant cardiovascular demand that are continued for longer or shorter periods of time. These activities may require training for between 4 and 6 h a day, 6 days a week; some examples are swimming, rowing, cycling, triathlon, and some long-distance running disciplines [5,6]. This type of exercise is related to biochemical-physiological adaptations with multiple manifestations, including cognitive-behavioral ones, aimed at restoring homeostasis and supercompensation [7]. In the search for these adaptations, gastro-intestinal symptoms (GIS) may also be encountered, which may include bloating, cramps, nausea, flatulence, abdominal pain, alteration of the bowel movement (urgency to defecate and/or diarrhea), leaky gut syndrome, regurgitation with regard to the “lower” symptoms, and belching, regurgitation, retrosternal heartburn, chest pain, and emesis with regard to the “higher symptoms” [5,8,9,10,11,12,13,14,15]. From a pathophysiological point of view, with the start of exercise, there is a redistribution of blood flow from the gastrointestinal tract towards the skeletal muscle and systems linked to ventilation and thermoregulation, with a potential picture of splachnic ischemia. This phenomenon becomes additive to that triggered by the elevation of catecholamines, which cause altered gastrointestinal motility, found in the alteration of the transit and absorption capacities of nutrients [5,10,11,12,13,14,15]. The sum of these effects causes an inflammatory condition of the digestive tract, potentially underlying permeability alteration, microbial translocation, and consequently systemic inflammation with mild endotoxinemia [16] and subsequent potential reperfusion injury [17]. In addition to being able to negatively influence training and competitions [18], these phenomena can have considerable relevance in the alteration of the intestinal microbiota (the complete set of microbes that exist naturally within a particular biological niche, approximately 500–1000 species, and their impact on human health) [19,20] and the immune system [21]. These issues are observed in reference to high-intensity physical activity, which shows negative effects both in terms of gastrointestinal problems and alteration of the intestinal microbiota. This is not a problem exclusive to professionals but also affects amateurs who, relative to their abilities, engage in high-intensity work. The aim of this article is to describe the main characteristics of these alterations, analyzing the possible role of substances in nutraceutical action.

## 2. Effects of Exercise: From the Digestive System to the Intestinal Microbiota

Analyzing the effects of exercise on the gut microbiota proves rather complex, as the literature presents a series of often conflicting and not entirely conclusive data [1,6,22,23,24]. A primary finding in animal models reported by Matsumoto et al. [25], followed by several other authors [26,27], is related to the increase in butyrate levels and butyrate-producing strains following physical activity. Despite these initial findings, numerous factors such as dietary, physiological, exercise-related, age-related, and the variety of the studied animal can complicate the analysis, leading to conflicting results. For example, some studies suggest an increase [25,28,29] or a decrease [26,30,31,32] in the Firmicutes to Bacteroidetes ratio due to exercise. Similarly, diverse results have been observed in humans. Analyzing physically active women (engaging in at least 3 h of exercise per week) compared with a sedentary control group, Bressa et al. [33,34] found a significant increase in butyrate producers, including *Faecalibacterium prausnitzii and Roseburia hominis*, along with increased levels of *Akkermansia muciniphila* and other bacteria with positive health effects such as *Bifidobacterium* spp., correlating with lower BMI and better metabolic health [35]. Analyzing professional rugby athletes, Clarke et al. [36] observed higher alpha-diversity, abundance in at least 40 microbial taxa, including *Akkermansia muciniphila*, and a reduction in *Bacteroides* and Lactobacilli compared with the control group (sedentary, lean subjects), confirming that athletes’ characteristics are also a factor linked to the composition of their microbiota [24]. Correlations between microbial composition and metabolic performance have also been noted, indicating an association between a high Firmicutes/Bacteroidetes ratio and VO2max [37], between microbial diversity, butyrate producers, and cardiorespiratory fitness, further highlighting peculiar microbial pathways in athletes involved in butyrate production, carbohydrate metabolism, and nitrogenous substances [37,38]. Analyzing sedentary, lean, and obese subjects undergoing a 6-week resistance training program (consisting of three sessions of 30–60 min of exercise per week) and dietary control revealed BMI-specific microbiota variations: increased *Faecalibacterium* in lean subjects and decreased in obese subjects, along with a reverse trend in *Bacteroides* [39,40]. Similarly, studying the effects on microbiota resulting from moderate-intensity exercise (3 times a week for 8 weeks) and whey protein consumption in overweight and obese men and women, an increase in bacterial diversity was found only when protein consumption was associated with exercise compared with protein consumption alone [41]. Particularly interesting are the results of Munukka et al. [42], who in a similar study reported that in overweight women previously sedentary, undergoing a six-week physical activity program, only half of the subjects examined showed an increase in the relative abundance of *Akkermansia muciniphila* and a decrease in Proteobacteria, along with a reduction in the abundance of several genes related to fructose and amino acid metabolism, defining an additional level of complexity. Consistent with these results, Castellanos et al. [23,39,43] highlighted that subjects adhering to the World Health Organization (WHO) guidelines for physical activity and diet showed increased microbial diversity associated with greater production of short-chain fatty acids (SCFA) compared with sedentary subjects. Exercise has been described by Mahdieh et al. [44] as capable of increasing Bifidobacteria levels in 18 obese women undergoing three weekly sessions of aerobic training lasting 30–45 min for 10 weeks, a very interesting finding considering the demonstration by Motiani et al. [45] of the modification of the microbial profile and the related reduction in exercise-induced intestinal inflammation in insulin-resistant subjects. In this context, exercise emerges as a factor capable of counteracting the progression of obesity while simultaneously promoting greater microbial diversity and a more balanced ratio between Firmicutes and Bacteroidetes bacteria in the gut [45,46]. It becomes interesting to correlate this data with that reported by Quiroga et al. [47] in obese boys, where training is associated with a reduction in Gammaproteobacteria and Proteobacteria levels and an increase in *Dialister*, *Blautia*, and *Roseburia levels*, with a profile more similar to that of normal-weight subjects and the decrease in NLRP3 signaling associated with obesity. Diversity has also been described in elderly subjects undergoing a training program compared with the counterparts of healthy sedentary subjects, with a more favorable *Bacteroides*/*Prevotella* ratio [48]. All the positive effects on the microbiota are summarized in Figure 1.

### 2.1. Exercise and Inflammation

Bonomini-Gnutzmann’s work summarizes the effects of exercise on the intestinal microbiota, considering parameters such as intensity and duration of exercise, helping to clarify the apparently dissonant picture found in the positive effects described so far and the possible negative pictures discussed in the introduction [1]. This 2022 work considered a total of 6277 studies relevant in terms of argument and subject, published in the last 5 years, of which 45 were isolated (6136 excluded after examination of title and abstract for lack of relevance and 95 excluded for duplication). Having examined these 45 with the established exclusion and inclusion criteria, 16 studies remained, of which seven had an observational design and nine were experimental, examining a total of 513 subjects. Of the studies examined, thirteen reported negative effects following aerobic training, including intestinal discomfort (one study) [49], an increase in intestinal fatty-acid binding protein (I-FABP, a marker of mucosal damage, three studies) [4,49,50], an increase in zonulin (two studies) [9,51] (a marker of increased intestinal permeability), and intestinal permeability (three studies) [4,9,50]. They also reported negative effects on the microbiota [51,52,53] with an increase in *Prevotella* spp. [54,55]. On the contrary, seven other studies report positive effects such as an increase in microbial counts [7,55,56] and biodiversity with the development of *Roseburia hominis*, *Bifidobacterium* spp., *Akkermansia muciniphila*, *Faecalibacterium prausnitzii* [19,33], *Coriobacteriaceae* [17], and an increase in the production of intestinal metabolites [39].

### 2.2. Exercise and Intestinal Permeability

The work carried out by Clark et al. [6] provides a notable overview of exercise-related disorders by analyzing the implications of the microbiota-gut-brain axis. The functionality of the intestinal barrier is guaranteed and regulated by the work of over 50 proteins that intervene in the regulation of endothelial and mucosal functionality, with particular reference to the tight junctions that intervene in the regulation of intestinal permeability, allowing the paracellular passage of desired molecules such as ions and water in leukocytes while controlling the translocation of microorganisms and the by-products of their metabolism [57,58]. This process is of fundamental importance for maintaining general and especially immune health [12]. The loosening of the protein structures of the tight junctions causes an increase in intestinal permeability, a condition defined as “leaky gut,” caused, among other things, by an excessive release of hormones related to stress (physical and/or psychological) [59]. The activation of the hypothalamic-pituitary-adrenal axis can in fact intervene in the stimulation of mast cells to release immune mediators such as histamine, proteases, and proinflammatory cytokines [60] by altering intestinal permeability [61]. This phenomenon allows bacterial lipopolysaccharide (LPS) to overcome the intestinal barrier, triggering immune and inflammatory responses. Once detected by CD14 and toll-like receptor 4 (TLR4), the release of tumor necrosis factor alpha (TNFα) occurs. interferon alpha (IFNα), interferon gamma (INFγ), interleukins (IL1β or IL6), and other proinflammatory cytokines [62], which contribute to further increasing intestinal permeability by acting on ZO1 and ZO2, generating patterns that can achieve endotoxinemia [62]. Among the factors related to exercise capable of causing gastrointestinal symptoms (which affect between 20 and 50% of athletes), intensity seems to play a decidedly relevant role [63], especially considering hyperthermia, ischemia, and hypoperfusion that may result. The resulting ischemia and hypoperfusion are stimuli capable of promoting further training of the tight junctions [12,64], up to the breakdown of the intestinal barrier with the related inflammatory response [65], and the increase in the production of reactive oxygen species (ROS) can further amplify intestinal permeability [62]. Various pieces of evidence published in the literature confirm this type of reaction. For instance, Jeukendrup et al., while analyzing 29 highly trained triathletes, found that as many as 93% experienced digestive disorders during the competition. This led, in two cases, to withdrawal [66] due to severe emesis and diarrhea. A further analysis carried out on young, healthy cyclists has highlighted how carrying out activities at 70% of the maximum for 4–10 h/week is already correlated to splacnic hypoperfusion, reduced gastrointestinal circulation, increased permeability, and lesions to the small intestine [67]. In agreement with these data, a further evaluation underlined how exercise at 70% of VO2max is sufficient to cause a 60–70% reduction in splanchnic blood flow, with ischemic phenomena and alteration of intestinal permeability, which were already starting to appear, for circulatory reductions of 50% [68]. Similarly, in a further evaluation of 18 ultradistance triathletes, mean plasma LPS concentrations increased from 0.081 to 0.294 ng/mL, and mean plasma anti-LPS immunoglobulin G concentrations decreased from 67.63 to 38.99 μg/mL [69]. Similar evaluations carried out in seven healthy cyclists also demonstrated a role for glucocorticoids, which released during intense exercise reduce the expression of TLRs and consequently the ability to produce anti-inflammatory cytokines and to manifest the immune response fully [70,71]. Prolonging exercise is a further factor capable of altering permeability, as demonstrated by the evaluation carried out on marathon runners who took more than 8 h to finish the route, where higher levels of endotoxins were found [72]. These studies demonstrate that the intensity and duration of exercise can cause increased intestinal permeability, followed by increased plasma LPS levels and potentially immunosuppression [67]. 

The main mechanisms of damage caused by intense and prolonged exercise are summarized in Figure 2.

### 2.3. Dietary Modulation of the Microbiota and Gastrointestinal Disorders: The Role of SCFA

The microbiota intervenes in the complex regulation of gastrointestinal function, in enteric immunity, in the regulation of some endocrine aspects, and in the management of oxidative stress [6,65,73,74]. To analyze these aspects, it is necessary to consider how, at the colon level, various polysaccharides and substances of vegetal origin are used as energy substrates and fermented by various microorganisms, including lactobacilli, *Bifidobacterium*, *Clostridium, and Bacteroides*, producing SCFA and gas that can be used by bacteria such as reductive acetogens, sulfate-reducers, and methanogens [6,75,76]. Among their various roles, SCFAs intervene in the regulation of colon pH, influencing the composition of the microbiota, intestinal motility and permeability, and epithelial proliferation [77]. Several factors can influence the type and extent of SCFA production, including microbiota composition, cross-feeding, quantity and type of host food intake, etc. [78,79,80]. Generally speaking, acetate, propionate, and N-butyrate are found in a molar ratio of 60:20:20 in the colon and feces [81]. An adequate presence of N-butyrate and propionate generally correlates with the implementation of the intestinal barrier and the reduction of inflammation [82], improving trans-epithelial resistance and preventing mucosal degradation, even if caused by hypoperfusion and ischemia such as those generated by intense and prolonged exercise [83], as well as constituting a primary energetic substrate for colonocytes [81]. N-Butyrate also intervenes in the modulation of neutrophil activity, the inflammatory response, and the structural proteins of tight junctions [83]. Current existing studies, as previously discussed, describe how intense exercise can influence SCFA production, which in turn impacts the hypothalamic-pituitary-adrenal axis and gastrointestinal health, promoting or compromising health and performance [84,85].

## 3. Acute, Chronic Exercise, and Microbiota

By analyzing the different effects of physical exercise on the intestinal microbiota, it is necessary to make a distinction between the effects observed acutely and the effects observed chronically, as systematically analyzed by Morh et al. [24].

### 3.1. Acute Effects of Physical Exercise on Microbiota

By analyzing the fecal metabolites and the microbiota of 20 amateur runners, Zhao et al. [86] found few variations at the level of alpha diversity contextual to variations in the abundance of some phylums (*Lentisphaerae* and acidobacteria), but above all at the species level, with an increase in Coriobacteriaceae and Succinivibrionaceae, implicated in the metabolism of bile salts, steroid hormones, and in the activation of various food-derived polyphenols [87], positively correlating with over 15 different metabolites. At the genus level, a reduction in the abundance of *Ezakiella*, *Romboutsia,* and *Actinobacillus* (a pathogen in the veterinary field and in periodontal diseases whose inhibition is potentially positive) was found at the same time as an increase in *Coprococcus* and *Ruminococcus bicirculans*. Scheiman et al. [88], analyzing the intestinal microbiota of subjects who would soon have participated in the Boston Marathon against the microbiota of sedentary subjects and collecting samples up to a week before and a week after the event, highlighted the increase in bacteria in the *Veillonella* genus after the marathon, finding its greater representation in runners compared with non-runners. Evidence is consistent with the ability of *Veillonella* to metabolize lactate [89]. These results were replicated by the same authors on a sample of ultramarathon runners and Olympic-level rowers analyzed before and after a training period [88]. It seems that the colonization of *Veillonella* can implement the Cori cycle with an alternative route for the metabolization of lactate (through the methylmallonin CoA pathway) [88], obtaining SCFAs that are released into circulation and used for the relevant metabolic purposes, potentially contributing to the athlete’s energy efficiency. High levels of intestinal lactate are related to the promotion of this mechanism. A third series of experiments where the same authors isolated a strain of *Veillonella atypica* from a stool sample of marathon runners and subsequently inoculated it into a mouse model highlighted a 13% increase in the time of the running test to exhaustion, with a significant reduction in inflammatory cytokine levels and more effective conversion of lactate to SCFA [88] compared with control. When considering these results, it is important to remember that in the mouse model studied, systemic lactate was able to cross the intestinal barrier, returning to the lumen to be converted to SCFA [88]. This study revealed how bacteria with identifiable positive potential in athletes can be efficiently and effectively transferred to improve performance. These results are partially overlapping with those of a case report that we previously described [90], whereby analyzing the microbiota of a 47-year-old Caucasian professional ultra-endurance athlete before and after an 800 km pedal ride plus 1200 km of running in 21 days (the Italy divide race held in July 2021), we found only a modest decrease in alpha diversity and a substantial stability of the Firmicutes/Bacteroidetes ratio, with pronounced variations only in *Prevotella* (+9.5%) and *Faecalibacterium* (+1.19%), as observed in other ultra-endurance athletes; instead, the most evident decline was observed in the taxa Lachnospiraceae (−269%) and *Sutterella* (−1.99%). The data seems to confirm that Mohr et al. found that the increase in *Veillonella*, absent in our case, is related to acute exercise while the increase in Bacteroidetes, as in our case, with a reduction of the Firmicutes/Bacteroidetes ratio, is related to chronic exercise, a situation probably representative of an athlete continuously in training or competition [40].

### 3.2. Chronic Effects of Physical Exercise on Microbiota

By analyzing the effects found in chronic conditions, Allen et al. demonstrated how training is able to modulate the composition and metabolic capacity of the microbiota [40] in previously sedentary subjects. Lean and obese subjects underwent 6 weeks of diet-controlled resistance training, followed by a 6-week washout. Exercise was correlated in lean subjects with changes in body composition and in obese subjects with an improvement in VO2max, regardless of diet. Particularly interesting are the changes at the level of butyrate producers, with particular reference to *Faecalibacterium* spp. and *Lachnospira* spp., with a greater impact on lean subjects compared with obese subjects. It is interesting to point out that after the washout, the changes at the microbiota level were largely lost, suggesting that the composition of the intestinal microbiota is related to the training condition, considering that the changes induced are mainly temporary and may require continuous stimuli [91]. Evaluations made on elderly subjects by Taniguchi et al. have demonstrated how resistance exercise intervenes in the modulation of the microbiota, with findings also on the cardiometabolic phenotype of the host. Analyzing 33 elderly men who underwent a 5-week training session showed that changes in alpha diversity negatively correlated with changes in systolic and diastolic blood pressure, especially during exercise. This is associated with a reduction in the abundance of *Clostridioides* and an increase in the abundance of *Oscillospira*, suggesting an association between variation at the taxa level and a reduction in cardiometabolic risk factors [92]. This data is very interesting to experts in light of what was published by Pluznick et al., who report how microbial SCFAs can act on blood pressure control by interacting with specific host receptor lines [93]. In a similar way, Morita et al., examining healthy elderly women subjected to a 12-week exercise program [94], described an increase in the relative abundance of *Bacteroides* for subjects who completed the exercise program, directly corresponding to the increase in test results of the 6 min walk. This correlates with data in the literature that attribute low levels of Bacteroides to an association with obesity and metabolic syndrome [95,96] in the context of healthy eating, while in a Western diet they can also correlate with a high BMI [97]. Analyzing prediabetic, overweight men subjected to an aerobic-anaerobic interval exercise program with 70-min sessions practiced three times a week, Liu et al. [98] found, despite global benefits on metabolism, a poor response from 30% of the subjects tested. The subjects who responded showed a notable declaration of insulin values and the HOMA index, highlighting a greater genetic expression of bacteria capable of producing SCFAs and metabolizing BCAAs. One of the longest studies carried out is that conducted by Kern et al., who considered the effects of regular aerobic training of different intensity but with similar energy expenditure continued for six months on the microbiota [99] on sedentary overweight subjects. In the different exercise groups compared with the control group, a change in beta diversity was found, with less heterogeneity and a greater increase in alpha diversity already after 3 months in the subjects who performed the most vigorous exercise. In the observation study by Keohane et al. [56] carried out on four athletes participating in a 33-day transoceanic crossing, it emerged that from the seventeenth day there was an increase in alpha diversity, regardless of cardiorespiratory changes, with a greater abundance of butyrate producers and species associated with better insulin sensitivity and metabolic greeting.

## 4. Sports-Specific Nutritional Approaches and Low-FODMAP Diets

In recent years, there has been a notable rise in participation in endurance and ultra-endurance sports such as triathlons and mountain races. However, a prevalent challenge in these activities is gastrointestinal discomfort, often exacerbated by the consumption of fats, fiber, proteins, or highly concentrated carbohydrate solutions during physical exertion. Running is one of the most strenuous forms of exercise on the gastrointestinal (GI) tract, with over 80% of athletes experiencing gas-trointestinal symptoms (GIS) during exercise. These symptoms result from physiological adaptations that occur during exercise, including reduced blood flow to the GI tract and alterations in gut motility and nutrient absorption. GIS includes both upper and lower GI discomfort, such as abdominal cramps, nausea, diarrhea, and rectal bleeding. Chronic GI disorders like irritable bowel syndrome (IBS), inflammatory bowel disease (IBD), and gastro-esophageal reflux disease (GERD) further complicate matters for athletes, affecting their quality of life and training consistency. Nutrition plays a crucial role in managing exercise-induced GIS and chronic GI conditions [15]. Although, in common understanding, the Mediterranean diet is considered a universally healthy approach due to the vast majority of applications [100], athletes often modify their diets before exercise to minimize GIS, with practices such as gluten-free or low-FODMAP diets being common. Research also suggests that some nutraceuticals and functional foods may help alleviate these gastrointestinal symptoms. Gastrointestinal symptoms can be influenced by several nutritional factors, so athletes may adhere to more stringent diets in an attempt to avoid aggravation of GIS. The most avoided foods are dairy, high-protein, and high-fiber foods, all in line with recommendations to limit protein, fat, and fiber consumption before exercise. More than half and one-third of runners with IBS/IBD and reflux reported avoiding dairy products before running, respectively [15]. Dairy products contain several nutrients that can potentially trigger GIS in this group of runners, including FODMAPs, protein, and fat. High-fiber foods are also avoided by runners with IBS/IBD and reflux before running. Dietary fiber can cause intestinal cramps, and the evidence on the effect of different types of fiber on IBS is complex. Finally, caffeine can negatively affect gastrointestinal function, and many runners avoid coffee or tea before running. A recent study [101] aimed to investigate the perceived effect of acute FODMAP intake on the severity of gastrointestinal symptoms. Although the clinical effectiveness of a low-FODMAP diet in treating IBS is established in the literature, research into the potential therapeutic effects in healthy athletes is limited. The main findings of the present study revealed that short-term low-FODMAP intake significantly improved exercise-related gastrointestinal symptoms in 69% of participants. From these results, it is inferred that both recreational and more trained athletes may benefit from short-term self-prescribed low-FODMAP approaches. This may have implications for longer-term FODMAP strategies during sustained training periods, which may provide additional nutritional support in maintaining training volume and/or intensity, especially in symptomatic individuals suffering from gastrointestinal disorders with exercise. These results could be due to a reduction in the amount of undigested carbohydrates that can be fermented in the intestines. Additional research is needed to explore the potential advantages of low-FODMAP diets for both recreational and trained athletes over prolonged periods of training.

## 5. The Possible Role of Probiotics

Research suggests that athletes and physically active individuals exhibit a more diverse array of gut microbes, including beneficial types like *Verrucomicrobia* and *Akkermansia*, compared with sedentary individuals. This reciprocal relationship between physical activity and gut bacteria suggests that during exercise, gut microbes metabolize indigestible carbohydrates into short-chain fatty acids (SCFA), serving as an energy source. Moreover, during intense anaerobic exercise, muscles produce lactate, which supports specific colon bacteria, potentially enhancing metabolic advantages and physical performance during workouts. Similarly, endurance training and antibiotic use may affect gut microbe diversity, with potential implications for overall health and exercise performance [102]. Physical exercise has recently been identified as another factor influencing the composition, diversity, and metabolic activity of the intestinal microbiota. However, the impact of physical exercise associated with dietary patterns and the type of training on the intestinal microbiota is not fully understood. The type of physical training and the athlete’s diet influence the relative abundance of intestinal microbiota at the genus and species levels. In bodybuilders, the highest levels of *Faecalibacterium*, *Sutterella*, *Clostridium*, *Haemophilus*, and *Eisenbergiella* were observed, while the lowest levels of *Bifidobacterium* and *Parasutterella* were noted, compared with long-distance runners [24].

The concept of probiotics has a rich history, dating back to Elie Metchnikoff’s suggestion in 1908 to manipulate the microbiota to replace harmful microbes with beneficial ones [103]. These live microorganisms offer numerous health benefits, including immune system support, protection of the intestinal barrier, and the production of beneficial substances like vitamins and SCFAs. Over recent decades, probiotic research has advanced significantly, leading to the identification and characterization of specific probiotic cultures. Today, a wide range of dietary supplements containing probiotics are commercially available, with many products targeting the health and performance of athletes. Several applications have taken into consideration the use of probiotics for the resolution of specific conditions, such as add-on therapy in the eradication of *Helicobacter pylori* [104], add-on therapy to conventional therapies for the treatment of diverticular disease [105], demonstrating significant potential in the management of various digestive problems, and also showing other innovative applications such as add-on therapy for respiratory diseases [106]. One of the most extensively studied strains due to its versatility is undoubtedly *Clostridium butyricum*, a gram-positive butyrate producer and currently the only cultivable and usable strain due to the volume of data proving its safety. *Clostridium butyricum* CBM588 exhibits remarkable beneficial capabilities attributed to the production of short-chain fatty acids (SCFAs), particularly butyric acid. It has long been used in some Eastern countries as an effective remedy for various gastrointestinal symptoms, such as intractable diarrhea and antibiotic-induced colitis. It is also extensively studied in the oncological field due to its ability to improve the health of the intestinal wall, enhance treatment tolerability, and reduce toxicity. Additionally, it can enhance host immunity and promote the growth of beneficial populations such as bifidobacteria [107,108]. Besides these specialized areas, probiotics can also be useful in other situations, such as those related to intestinal disorders induced by sports. The use of probiotic microorganisms, especially those within the gram-positive *Lactobacillus* and *Bifidobacterium* genera, has been shown to influence the gut microbiota and could offer an extra nutritional approach in instances of acute gastrointestinal (GI) disruption, such as gastro-intestinal dysbiosis and exercise-induced GI permeability. Athletes, who undergo intense and prolonged physical exertion, may benefit from probiotic use as it can enhance immune defenses and exercise adaptation. Additionally, the intestinal microbiota may indirectly influence athletic performance and post-training recovery by regulating the immune response and modulating gene expression associated with immune cell activity. However, further research is needed to fully understand the effects of probiotics on athletic performance and overall health. Gastrointestinal dysfunction can compromise nutrient absorption, cause gastrointestinal symptoms, and reduce performance. Supplementation with probiotics, in combination with other dietary strategies, may assist athletes with gastrointestinal issues, improve gut health, and potentially enhance athletic performance. While some studies have reported positive effects, such as a reduction in intestinal permeability and gastrointestinal symptoms, others have not found significant benefits. Additionally, the duration and type of probiotic supplementation can influence study outcomes, making it challenging to draw definitive conclusions about the effectiveness of probiotics in improving gut health and athletic performance.

One significant challenge in this area of research is the relatively low prevalence of gastrointestinal diseases overall, making it challenging to conduct studies with large participant pools. Nonetheless, athletes with gastrointestinal issues may benefit from probiotic supplementation alongside other dietary approaches. Moreover, probiotics have the potential to enhance gut health, offering various indirect advantages for athletes. A recent study [109] aimed to explore the effects of multiple doses of probiotics on athletic performance and gastrointestinal health during physical activity. Participants who received a probiotic capsule containing strains of *Lactobacillus*, *Bifidobacterium*, and *Streptococcus* demonstrated a notable improvement in running endurance compared with those who received a placebo. Additionally, probiotic supplementation resulted in a modest reduction in gastrointestinal discomfort and changes in serum lipopolysaccharide levels, indicating potential benefits for both the immune system and gut health during exercise. These findings suggest that probiotics may play a beneficial role in enhancing athletic performance and overall well-being among active individuals [109].

## 6. Nutraceuticals Potentially Usable in Intestinal Well-Being

Various substances with nutraceutical action can be used, not only by virtue of the properties for which they have been classically studied but also with a view to remodulating the microbiota and for prebiotic purposes [110]. The prebiotic approach, in the perspective of managing the described issues, should not be handled in a “one-size-fits-all” manner, as it has been done for several years with various types of fibers, but rather should involve the use of substances with as specific action as possible to plan goal-oriented applications. Below is an analysis of some of the main applications.

### 6.1. Berberine

Berberine (BBR) is a compound extracted from various medicinal plants and is used to treat a wide range of diseases, including tumors, cardiovascular diseases, and digestive issues. Studies have shown that BBR has numerous beneficial effects, including promoting insulin secretion, reducing lipid accumulation, and modulating the immune system. Recently, there has been an increased focus on the role of the gut microbiota (GM) in health and disease.

It reduces GM diversity and shifts the balance of bacterial species, including *Desulfovibrio* spp., *Eubacterium* spp., and *Bacteroides* spp. Studies reveal that while *Bacteroides* spp. thrive in the colon and terminal ileum after BBR treatment, populations of *Ruminococcus gnavus*, *Ruminococcus schinkii*, *Lactobacillus acidophilus*, *Lactobacillus murinus*, and *Lactococcus lactis* decrease. Some studies indicate that berberine can increase levels of *Akkermansia muciniphila* in the intestinal microbiota [103,109,111,112,113,114]. BBR also modulates immune cell activity, suppressing the expression of inflammatory factors such as interleukin (IL)-1β, IL-4, IL-10, and tumor necrosis factor (TNF)-α, thereby mitigating low-grade inflammation. Furthermore, berberine alters intestinal bacterial populations by inhibiting bile salt hydrolase (BSH) activity and promoting the synthesis of butyrate, a short-chain fatty acid beneficial for metabolic health. Several clinical evaluations have demonstrated the effectiveness of BBR in the management of diarrhea in subjects with functional gastrointestinal disorders [115]. Additionally, BBR regulates the intestinal microbiota through tryptophan metabolism and activation of the tryptophan receptor (AhR), improving the damaged intestinal barrier [114]. These effects underscore the potential of BBR in reducing intestinal inflammation.

### 6.2. Curcumine

Curcumin, the primary curcuminoid isolated from *Curcuma longa* L., is renowned for its antioxidant, anti-inflammatory, and anti-cancer properties. What makes curcumin particularly intriguing is its metabolism through digestion by intestinal microbiota, leading to the formation of bioactive metabolites in the intestine. This natural phenolic compound finds applications in various industries, such as food, textiles, and pharmaceuticals. Thanks to the action of enzymes produced by intestinal bacteria, curcumin undergoes various metabolic modifications, including reduction, demethylation, hydroxylation, and acetylation. Several bacterial species are involved in this process, including *Escherichia coli*, *Blautia* spp., *Bifidobacterium longum*, and others. It is interesting to note that some curcumin metabolites, such as tetrahydrocurcumin, may exhibit similar or even superior potency compared with curcumin itself, offering potential therapeutic benefits such as neutralizing free radicals and inhibiting inflammatory cytokines. Despite its broad therapeutic potential, curcumin’s bioavailability is notably low, necessitating pharmaceutical technologies for enhancement. Extensive research supports curcumin’s beneficial effects on conditions such as cancer, diabetes, autoimmune disorders, and neurodegenerative diseases. Studies indicate that curcumin can positively influence the composition of the gut microbiota. In a human trial, oral curcumin supplementation resulted in significant alterations in the gut microbiota compared with the placebo, with increased levels of certain bacterial species, including *Clostridium*, *Bacteroides*, and *Citrobacter*, and a decreased abundance of *Blautia* and *Ruminococcus*. In rats fed a high-fat diet, curcumin countered the increase in *Ruminococcus* species linked to diabetes and inflammation. Additionally, curcumin supplementation in rats reduced the abundance of families associated with systemic diseases while increasing beneficial bacteria like bifidobacteria and lactobacilli, known for their anti-tumoral properties. In a mouse model of colorectal cancer, curcumin treatment reduced the levels of cancer-related microbial species like *Prevotella* and *Ruminococcus*. Curcumin has also been shown to inhibit the activation of NF-κB, involved in the production of pro-inflammatory cytokines in IBD, by acting on the TLR-4 receptor and reducing the degradation of IκB protein, thus inhibiting the inflammatory cascade and reducing damage to the intestinal mucosa. Finally, curcumin may act through the activation of the peroxisome proliferator-activated receptor γ (PPAR-γ), which plays a role in inhibiting inflammation in the colon [116]. Overall, curcumin represents an intriguing therapeutic option that deserves further investigation and clinical studies to confirm its beneficial effects in clinical practice. Overall, current evidence suggests curcumin’s protective effect by promoting a shift from pathogenic to beneficial bacterial strains in the gut [117,118,119].

### 6.3. Green Tea

Green tea, derived from the leaves of the *Camellia sinensis* Kuntze plant, is deeply rooted in traditional Chinese medicine and renowed for its therapeutic benefits attributed to polyphenolic compounds. These compounds exhibit diverse helath-promoting properties, including antioxidant, anti-aging, and anti-inflammatory effects. Daily consumption of green tea appears to reduce the risk of inflammatory bowel disease (IBD). Epigallocatechin gallate (EGCG), a component of green tea, modulates the composition of the intestinal microbiota and its metabolites, increasing the presence of short-chain fatty acid (SCFA)-producing bacteria such as *Akkermansia* and enhancing SCFA production [120]. This contributes to promoting an anti-inflammatory and antioxidative state in the intestine. Human studies have shown that regular consumption of green tea increases the prevalence of certain bacterial families in the gut, such as Firmicutes and Actinobacteria, while simultaneously reducing the abundance of Bacteroidetes. Notably, green tea promotes beneficial bacteria like *Roseburia* spp., *Faecalibacterium* spp., and *Bifidobacterium longum*, known for their production of short-chain fatty acids, while suppressing certain species within the *Prevotella* genus. Experiments on mouse models have demonstrated that green tea polyphenols can elevate the population of *Faecalibacterium prausnitzii*, correlating with mitigated weight gain and reduced inflammation in the intestines and liver induced by a high-fat diet. In mice fed a high-fat diet and supplemented with green tea extract, there was an increase in the abundance of Bacteroidetes and Oscillospira spp. families, alongside a decrease in the *Peptostreptococcaceae* family. Moreover, a notable reduction in the Firmicutes/Bacteroidetes ratio was observed, accompanied by a positive regulation of *Akkermansia muciniphila*. In vitro studies have highlighted the ability of green tea polyphenols to modulate the composition of the intestinal microbiota, suppressing pathogenic bacteria such as *Clostridium perfringens*, *Clostridium difficile*, and various *Bacteroides* species while enhancing beneficial strains like *Lactobacillus* and *Bifidobacterium*. Collectively, these findings suggest that green tea holds promise as a potential agent against obesity through its intricate interactions with the gut microbiome.

### 6.4. Boswellia

Boswellia serrata, commonly known as frankincense, has been revered for centuries for its therapeutic properties, particularly as a natural anti-inflammatory, immunomodulatory, and antimicrobial agent. One of its key bioactive components, 3-O-acetyl-11-keto-β-boswellic acid (AKBA), has garnered significant interest for its potential health benefits [118]. However, the impact of AKBA on the gut microbiome and blood metabolites remains largely unexplored. Understanding how AKBA influences these physiological parameters could provide valuable insights into its potential therapeutic applications. AKBA was tested on male and female mice, demonstrating a significant reduction in gut bacterial richness in male mice but no effect on females. However, both male and female mice showed an increase in *Akkermansia muciniphila*, and in females, also in *Bifidobacterium* [121]. These results suggest a potential prebiotic effect of AKBA, which could positively influence the composition of the gut microbiota. Additionally, Giacosa et al. investigated how supplementation with extracts of *Curcuma longa* and *Boswellia serrata* during a low FODMAP regimen is associated with a significant decrease in abdominal bloating in subjects with IBS and small bowel dysbiosis [122,123,124]. Further studies are needed to fully understand the mechanism of action of Boswellia serrata and its potential impact on intestinal health.

### 6.5. Cranberry

The cranberry, also known as the Vaccinium berry, has garnered attention from the scientific community in recent years for its potential beneficial properties for human health. This fruit has been the subject of numerous studies that have highlighted its role in modulating the function of the intestinal microbiota and reducing cardiometabolic risk factors.

Cranberries, rich in polyphenols, have gained attention in scientific research for their potential health benefits. These bioactive compounds, including proanthocyanidins (PACs), play a crucial role as antioxidants and inflammation modulators. An intriguing aspect is the ability of cranberry polyphenols to influence the human intestinal microbiota, which plays a fundamental role in intestinal health and overall metabolism. Cranberry polyphenols, not fully absorbed in the small intestine, reach the colon, where they are metabolized by intestinal bacteria into bioactive compounds, including organic acids. These metabolites, along with cranberries themselves, influence the composition of the intestinal microbiota, promoting the growth of beneficial bacteria such as *Lactobacillus*, *Bifidobacterium*, and *Akkermansia muciniphila* [125]. Modulating the intestinal microbiota is associated with various health benefits, including improved weight management, reduced inflammation, and enhanced protection against metabolic disorders like obesity and diabetes. In a recent study conducted on healthy adults [126,127], consumption of freeze-dried cranberry powder led to a significant reduction in the abundance of Firmicutes bacteria, accompanied by an increase in *Bacteroidetes*. This change in microbiota composition may have important implications for health, as the reduction in Firmicutes has been associated with better weight and metabolic health. Furthermore, consumption of cranberry powder was shown to positively influence the production of short-chain fatty acids (SCFAs), important for intestinal and energy metabolism health. Cranberry polyphenols, in addition to influencing the composition of the intestinal microbiota, can also modulate gene expression at the intestinal level through their action on microRNAs (miRNAs) [125]. This complex interaction between cranberry polyphenols, the intestinal microbiota, and miRNAs may contribute to the beneficial effects of these fruits on health. However, further research is needed to fully understand the mechanisms involved and to identify new biomarkers that can be used to monitor the effectiveness of the cranberry intervention.

## 7. Conclusions

Sport-related gastrointestinal disorders are particularly common in athletes, both during training and competitions. They encompass a wide range of symptoms associated with gastrointestinal distress, such as diarrhea, nausea, vomiting, heartburn, belching, and so forth. These gastrointestinal disturbances appear to be very common, with the severity and number of gastrointestinal symptoms seeming to depend on the intensity, volume of physical exercise, and sport practiced, to the extent that some studies report a documented prevalence exceeding 80%. Athletes are usually at a higher risk compared with amateurs; however, even in the latter case, engaging in high-intensity activities can lead to the development of such issues.

Despite the often sparse and conflicting evidence on sport-related gastrointestinal disorders, in this narrative analysis, we have attempted to clarify the causes, consequences, and a series of possible solutions, both dietary and nutraceutical. Indeed Physical exercise can cause acute and chronic intestinal alterations, which also seem to be mediated by modulation of the intestinal microbiota. Acute alterations have a transient nature and, from an interpretative point of view, could represent adaptations capable of ensuring an improvement in the utilization and management of energy resources in athletes; in this context, the increase in *Veilonella* recorded in some studies, by modulating the activity of the Cori Cycle, could provide an alternative and additional way to metabolize lactate, allowing further production of SCFAs capable of improving VO2max with undoubted advantages on presentation. In the chronic phase—although data are scarce—there is a reduction in *Bacteroides*, resulting in an increase in bacteria capable of producing SCFAs and metabolizing BCAAs, generating positive effects from a metabolic point of view.

The positive metabolic alterations determined by modulations of the intestinal microbiota induced by physical exercise, however, contrast with the onset of gastrointestinal symptoms that can compromise the athlete’s performance and health. Therefore, sports medicine clinicians must be able to recognize and treat them correctly by identifying specific therapeutic and/or nutritional strategies. By modulating the microbiota through nutritional and nutraceutical strategies, it may be possible to reduce gastrointestinal-related issues.

Although the Mediterranean diet is universally considered a healthy approach, the high fiber intake could worsen symptoms in some cases in athletes with sport-related gastrointestinal disorders. Guidelines suggest reducing fat and fiber intake before a sporting event; however, many athletes perceive improvements by reducing dairy products. In recent years, the low-FODMAPs diet has been gaining ground as a nutritional treatment for IBS and IBD, which involves reducing the consumption of a wide range of foods containing fermentable oligosaccharides, disaccharides, monosaccharides, and polyols, which are short-chain carbohydrates. Although research on the therapeutic effects and potential effects of a low-FODMAPs diet on athletes is limited, this nutritional approach seems to correlate in one study conducted with a significant improvement in gastrointestinal symptoms in about two-thirds of the participating athletes.

Some nutraceuticals, on the other hand, are increasingly attracting interest by providing positive metabolic effects and exerting a direct and indirect prebiotic action that can be useful for the athlete’s performance and gastrointestinal health. Specifically, berberine, in some studies, is associated with an increase in *Akkermansia muciniphila* and a reduction in a series of pro-inflammatory markers such as interleukin (IL)-1β, IL-4, IL-10, and tumor necrosis factor (TNF)-α, ensuring an improvement in intestinal permeability. Instead, curcumin, the main curcuminoid extracted from the rhizomes of Curcuma longa L., can exert anti-inflammatory and immunomodulatory effects by inhibiting inflammatory mediators and cytokines, scavenging oxygen free radicals, and other processes, potentially useful in sport-related gastrointestinal disorders and in the treatment of IBS and IBD. The consumption of green tea seems to be associated with a reduction in the risk of IBS and IBD; the therapeutic potential seems to derive from EGCG. It is associated with a positive modulation of the intestinal microbiota, increasing bacteria that produce short-chain fatty acids (SCFA) like *Akkermansia muciniphila*, potentially improving energy resource management in performance and intestinal permeability. Boswellia serrata is recognized for its undisputed anti-inflammatory potential; it is capable, in fact, of interfering with the activity of cyclooxygenase 2 (COX-2), interfering with the production of prostaglandin E2 (PGE2). Another component to mention is the polyphenols of cranberry (particularly PACs); they reach the terminal parts of the intestine, where they are metabolized into organic acids capable of promoting the growth of beneficial bacteria such as *Lactobacillus, Bifidobacterium, and Akkermansia muciniphila*, improving the production of SCFAs.

In conclusion, we can state that moderate physical activity also benefits the intestinal microbiota, while high-intensity physical activity can cause severe issues. In this regard, the use of targeted nutraceuticals, applied with specific objectives, can constitute a valid preventive approach and, if needed, be directed towards the treatment and management of these situations.

Further studies are needed to verify the causes, the complex etiology, and the incidence of sport-related gastrointestinal disorders in athletes. Scientific works are also needed, possibly conducted under suitable conditions and on a large number of athletes, to investigate the efficacy profile of certain nutritional strategies and nutraceuticals that seem to be able to improve intestinal symptoms by positively modulating the intestinal microbiota.

## Figures and Tables

**Figure 1 microorganisms-12-00804-f001:**
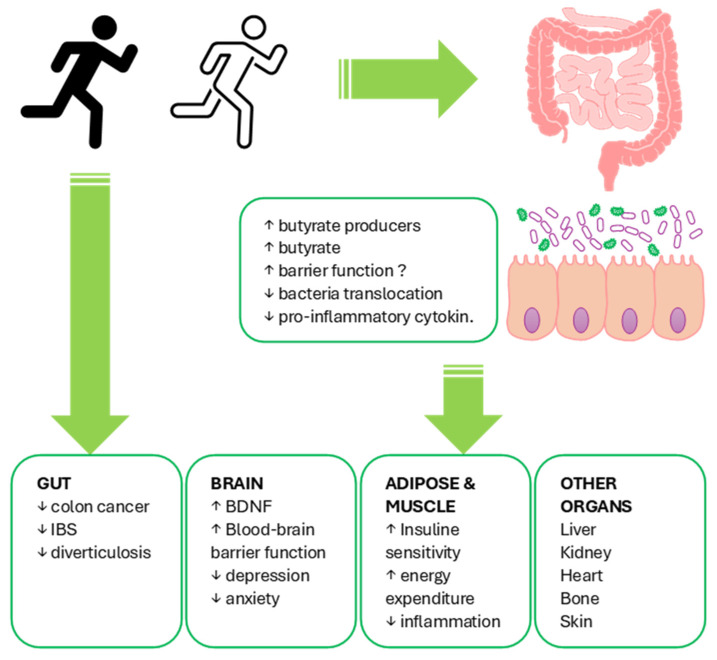
Summary of the potential positive effects of exercise on intestinal health and the related repercussions on other organs and systems [22].

**Figure 2 microorganisms-12-00804-f002:**
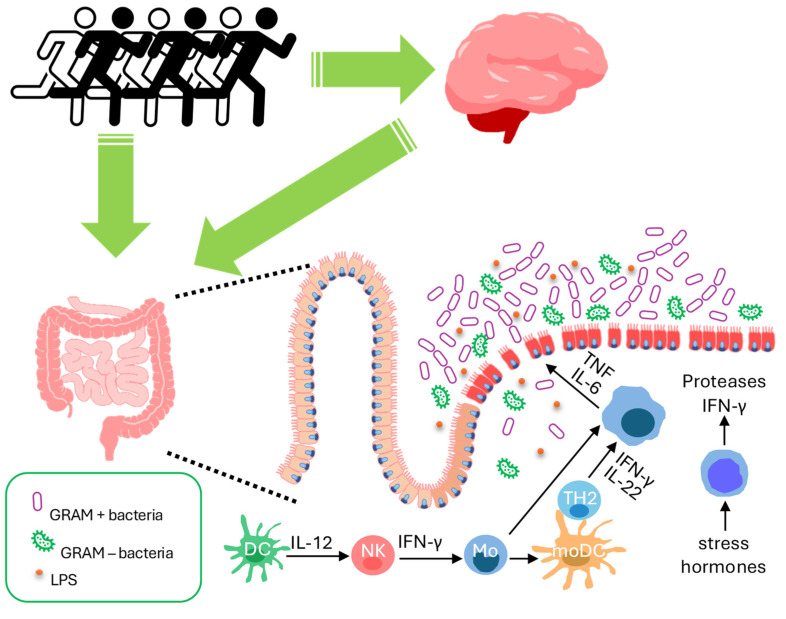
Summary of the main mechanisms of intestinal damage caused by intense and prolonged physical exercise: thermal stress, splachnic ischemia, oxidative stress, activation of the hypothalamic-pituitary-adrenal axis, inflammatory reactions, and immune alteration (see text) [6].

## Data Availability

Not applicable.

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
