# Peer review of "Sports-Related Gastrointestinal Disorders: From the Microbiota to the Possible Role of Nutraceuticals, a Narrative Analysis"

_microorganisms, 2024, doi:10.3390/microorganisms12040804_

Round 1

Reviewer 1 Report

Comments and Suggestions for Authors

microorganisms-2951393-peer-review-v1

This is an interesting paper, putting on the focus changes in the human microbiota as consequences of active sport and occasional exercises and possible consequences. Moreover, the role of probiotics in controlling the consequences of sport practices on GIT microbiota were suggested. However, in my opinion, the role of the probiotics can maybe be extended, and if there are some relevant examples and evidence for the possible benefits, to be presented in the current manuscript.

I would like to suggest that paper can be considered to be accepted by the Editor, however, some adjustments, corrections, and clarification will need to be considered by the authors.

Ln75: In my opinion will be correct if the reference number will be mentioned after Bressa et al.

Ln80. Regarding changes in taxonomy of lactobacilli from 2020, in this context will be more appropriate to use English word "lactobacilli" as was suggested by authors of taxonomy change from 2020, so cannot confuse with sensu strictu name of Lactobacillus, that represent only small part of previous genus (before changes)

Ln94: maybe add reference [42] after Munukka et al., .

Ln98: add reference 98 after Castella et al. . This will be all 3 cited references.

Please, adjust entire manuscript for the similar issues.

Ln106: please, be clearer. In the sentence "In this context, exercise can be recognized as a factor capable of stabilizing the progression of 106 obesity, " is exercise stopping or promoting obesity? As whiten is not very clear.

On figure 1, please, provide meaning of having one black and one white runner.

Ln131: Prevotella needs to be in italics. And correct to "spp."

Ln133-134: Bacterial names need to be in italics.

Ln162-165: Please, adjust structure of this sentence.

ln173:38.99 μg/ml

Ln194: please, use italics for bacterial species and consider to change to English word "lactobacilli"

Ln213: Morh et al.

Please, be sure that Latin bacterial names are in italics. Please, check entire manuscript for this kind of adjustments.

ln274: al. needs to be with not capital a.

Ln350: ... ones [103].

Well, Metchnikoff suggested that the use of some LAB as beneficial and suggested that they can have beneficial role, but concept of probiotics is coming a bit later.

Is this reference (103) appropriate?

Ln357: Helicobacter pylori needs to be in italics. Is Helicobacter pylori really related to sport practice individuals? Maybe write a bit more, and maybe provide different examples. Please, try to use appropriate examples direct link to the topic.

Ln383: What Streptococcus? This genus is very specific, since only 3 species are considered as safe and all other are pathogens. Please, be more specific what species were applied as probiotics in these cases.

Lm390: do you will talk about prebiotics?

for the different plans, is always suggested to provide Latin names, with aim to avoid misunderstandings related to some variations in English names.

for all species, previously been part of genus Lactobacillus, please, be sure that you will use appropriate names according to teh changes from 2020 and when appropriate abbreviated according to the recommendations from 2023. (https://doi.org/10.1099/ijsem.0.004107.32293557 https://doi.org/10.1163/18762891-20230114).

Ln406: you have introduced abbreviation (BBR), please, use it.. It is really needed to have this abbreviation.

Ln438: If you use word (bifidobacteria and lactobacilli) in English, then do not need to be in italics.

Maybe at the end a 2 lines clear take home message need to be added.

Author Response

This is an interesting paper, putting on the focus changes in the human microbiota as consequences of active sport and occasional exercises and possible consequences. Moreover, the role of probiotics in controlling the consequences of sport practices on GIT microbiota were suggested. However, in my opinion, the role of the probiotics can maybe be extended, and if there are some relevant examples and evidence for the possible benefits, to be presented in the current manuscript.

Taking into consideration these observations, further information regarding the role of probiotics has been added; specifically, the role of a widely used probiotic in gastrointestinal and other disorders has been described, highlighting the potential benefits.

The added text is between lines 363 and 372: “One of the most extensively studied strains due to its versatility is undoubtedly Clostridium butyricum, Gram-positive butyrate producer and currently the only cultivable and usable strain due to the volume of data proving its safety. Clostridium butyricum CBM588 exhibits remarkable beneficial capabilities attributed to the production of short-chain fatty acids (SCFAs), particularly butyric acid. It has long been used in some Eastern countries as an effective remedy for various gastrointestinal symptoms, such as intractable diarrhea and antibiotic-induced colitis. It is also extensively studied in the oncological field due to its ability to improve the health of the intestinal wall, enhancing treatment tolerability and reducing their toxicity. Additionally, it can enhance host immunity and promote the growth of beneficial populations such as bifidobacteria." Thanks for the suggestion.

I would like to suggest that paper can be considered to be accepted by the Editor, however, some adjustments, corrections, and clarification will need to be considered by the authors.

Ln75: In my opinion will be correct if the reference number will be mentioned after Bressa et al.

Updated, the reference number has been included after Bressa et al. Thanks for the correction. (line 78)

Ln80. Regarding changes in taxonomy of lactobacilli from 2020, in this context will be more appropriate to use English word "lactobacilli" as was suggested by authors of taxonomy change from 2020, so cannot confuse with sensu strictu name of Lactobacillus, that represent only small part of previous genus (before changes)

Noted, thanks for the clarification. (line 83)

Ln94: maybe add reference [42] after Munukka et al., .

Sure, the modification has been made. Thanks for the suggestion. (line 97)

Ln98: add reference 98 after Castella et al. . This will be all 3 cited references.

Please, adjust entire manuscript for the similar issues.

The reference has been added after Castella et al. Similar adjustments will be made throughout the entire manuscript. (lines 102)

Ln106: please, be clearer. In the sentence "In this context, exercise can be recognized as a factor capable of stabilizing the progression of 106 obesity, " is exercise stopping or promoting obesity? As whiten is not very clear.

On figure 1, please, provide meaning of having one black and one white runner.

The previous sentence "In this context, exercise can be recognized as a factor capable of stabilizing the progression of obesity, promoting microbial diversity, and a better Firmicutes/Bacteroidetes ratio [45,46]" has been replaced with the new one provided: “In this context, exercise emerges as a factor capable of counteracting the progression of obesity, while simultaneously promoting greater microbial diversity and a more balanced ratio between Firmicutes and Bacteroidetes bacteria in the gut [45,46].” (lines 109-111)

The inclusion of white and black figures was solely intended to convey movement and intensity to the scene, without any additional connotation.

Ln131: Prevotella needs to be in italics. And correct to "spp."

The correction regarding the italicization of "Prevotella" and the adjustment to "spp." in line 131 has been noted. Thanks for the correction. (line 135)

Ln133-134: Bacterial names need to be in italics.

Exactly, thank for the correction. (lines 137-138)

Ln162-165: Please, adjust structure of this sentence.

The previous sentence “Various evidence published in the literature confirms this type of reaction, Jeukendrup et.al analyzing 29 highly trained triathletes found that as many as 93% experienced digestive disorders during the competition, leading in 2 cases to withdrawal [66] due to severe emesis and diarrhea.” has been replaced with the new one provided: “Various evidence published in the literature confirms this type of reaction. For instance, Jeukendrup et al., while analyzing 29 highly trained triathletes, found that as many as 93% experienced digestive disorders during the competition. This led, in 2 cases, to withdrawal [66] due to severe emesis and diarrhea." (lines 166-169)

ln173:38.99 μg/ml

Ok, thanks for the correction. (line 177)

Ln194: please, use italics for bacterial species and consider to change to English word "lactobacilli"

Ok, thanks for the correction. (line 198)

Ln213: Morh et al.

Please, be sure that Latin bacterial names are in italics. Please, check entire manuscript for this kind of adjustments.

Understood. It will be ensured that Latin bacterial names are italicized throughout the entire manuscript for consistency. Thank you for bringing this to our attention.

ln274: al. needs to be with not capital a.

Ok, thanks for the correction. (line 277)

Ln350: ... ones [103].

Well, Metchnikoff suggested that the use of some LAB as beneficial and suggested that they can have beneficial role, but concept of probiotics is coming a bit later.

Is this reference (103) appropriate?

Ok, thanks for the correction. (line 353)

Yes, it is appropriate to include reference (103), as Metchnikoff indeed suggested over 100 years ago the possibility of using beneficial bacteria to replace harmful ones, as indicated in the ISSN guidelines.

Ln357: Helicobacter pylori needs to be in italics. Is Helicobacter pylori really related to sport practice individuals? Maybe write a bit more, and maybe provide different examples. Please, try to use appropriate examples direct link to the topic.

Thanks for the correction.

Regarding Helicobacter pylori, the intention was to provide an overview of the various specific applications of probiotics. (line 360) However, a brief sentence has been added to clarify the intent.

Ln383: What Streptococcus? This genus is very specific, since only 3 species are considered as safe and all other are pathogens. Please, be more specific what species were applied as probiotics in these cases.

In the mentioned context, only strains of Streptococcus considered safe have been employed. (line 397)

Lm390: do you will talk about prebiotics?

for the different plans, is always suggested to provide Latin names, with aim to avoid misunderstandings related to some variations in English names.

for all species, previously been part of genus Lactobacillus, please, be sure that you will use appropriate names according to teh changes from 2020 and when appropriate abbreviated according to the recommendations from 2023. (https://doi.org/10.1099/ijsem.0.004107.32293557https://doi.org/10.1163/18762891-20230114).

In the article, there will be no discussion of prebiotics in general, but it is emphasized how many substances may be able to modulate the gut microbiota, with prebiotic implications. In this regard, a few lines (407-410) have been added to complete the introduction to the paragraph: “The prebiotic approach, in the perspective of managing the described issues, should not be handled in a "one-size-fits-all" manner as it has been done for several years with various types of fibers, but rather should involve the use of substances with as specific action as possible, to plan goal-oriented applications.”

Ln406: you have introduced abbreviation (BBR), please, use it.. It is really needed to have this abbreviation.

Ok, thanks for the correction.

Ln438: If you use word (bifidobacteria and lactobacilli) in English, then do not need to be in italics.

Maybe at the end a 2 lines clear take home message need to be added.

Thanks for the correction (line 455) and suggestion.

Regarding the suggestion, a summary has been included in the conclusion to make the message clearer: “In conclusion, we can state that moderate physical activity also benefits the intestinal microbiota, while high-intensity physical activity can cause severe issues. In this regard, the use of targeted nutraceuticals, applied with specific objectives, can constitute a valid preventive approach and, if needed, be directed towards the treatment and management of these situations.” (lines 597-600)

Reviewer 2 Report

Comments and Suggestions for Authors

In the present narrative review Bertuccioli et al analyzed the relationship between intense physical exercise and gastrointestinal disorders, with a focus on the role of gut microbiota on pathogenesis, as well as on possible modulation to treat such condition by probiotics and nutraceuticals.

My only main comment is that Authors should better contextualize the problem. Indeed professional athletes and amateurs are different settings as I suppose that professionals may have better cardiovascular conditions/improved splanchnic circulation that may protect against transitory intestinal ischemia due to exercise. Such data indeed, seem to be in contrast to the fact that exercise may have a beneficial effect on microbiota composition: therefore, what should be the main message, that moderate training should be preferred?

Minor comment. Figure 1: the “other organs” panel is quite vague.

Author Response

In the present narrative review Bertuccioli et al analyzed the relationship between intense physical exercise and gastrointestinal disorders, with a focus on the role of gut microbiota on pathogenesis, as well as on possible modulation to treat such condition by probiotics and nutraceuticals.

My only main comment is that Authors should better contextualize the problem. Indeed professional athletes and amateurs are different settings as I suppose that professionals may have better cardiovascular conditions/improved splanchnic circulation that may protect against transitory intestinal ischemia due to exercise. Such data indeed, seem to be in contrast to the fact that exercise may have a beneficial effect on microbiota composition: therefore, what should be the main message, that moderate training should be preferred?

Thanks for the suggestion. The message has been clarified both in the conclusions and in the introduction with short texts.

In the introduction, the following text has been added: “These issues are observed in reference to high-intensity physical activity, which shows negative effects both in terms of gastrointestinal problems and alteration of the intestinal microbiota. This is not a problem exclusive to professionals but also affects amateurs who, relative to their abilities, engage in high-intensity work.” (lines 62-65)

In the conclusions, the following text has been added: “Athletes are usually at a higher risk compared to amateurs; however, even in the latter case, engaging in high-intensity activities can lead to the development of such issues.” (lines 543-545)

Minor comment. Figure 1: the “other organs” panel is quite vague.

Thanks for pointing that out. The specific organs we are primarily referring to are detailed below.